# A phase I/II study of triple-mutated oncolytic herpes virus G47Δ in patients with progressive glioblastoma

Tomoki Todo [1,2]✉, Yasushi Ino [1,2], Hiroshi Ohtsu [3,4], Junji Shibahara[5] & Minoru Tanaka [1,2]

Here, we report the results of a phase I/II, single-arm study (UMIN-CTR Clinical Trial Registry UMIN000002661) assessing the safety (primary endpoint) of G47Δ, a triple-mutated oncolytic herpes simplex virus type 1, in Japanese adults with recurrent/progressive glioblastoma despite radiation and temozolomide therapies. G47Δ was administered intratumorally at $3 \times 10^8$ pfu (low dose) or $1 \times 10^9$ pfu (set dose), twice to identical coordinates within 5–14 days. Thirteen patients completed treatment (low dose, $n = 3$; set dose, $n = 10$). Adverse events occurred in 12/13 patients. The most common G47Δ-related adverse events were fever, headache and vomiting. Secondary endpoint was the efficacy. Median overall survival was 7.3 (95%CI 6.2–15.2) months and the 1-year survival rate was 38.5%, both from the last G47Δ administration. Median progression-free survival was 8 (95%CI 7–34) days from the last G47Δ administration, mainly due to immediate enlargement of the contrast-enhanced area of the target lesion on MRI. Three patients survived >46 months. One complete response (low dose) and one partial response (set dose) were seen at 2 years. Based on biopsies, post-administration MRI features (injection site contrast-enhancement clearing and entire tumor enlargement) likely reflected tumor cell destruction via viral replication and lymphocyte infiltration towards tumor cells, the latter suggesting the mechanism for "immunoprogression" characteristic to this therapy. This study shows that G47Δ is safe for treating recurrent/progressive glioblastoma and warrants further clinical development.

[1] Division of Innovative Cancer Therapy, Advanced Clinical Research Center, and Department of Surgical Neuro-Oncology, The Institute of Medical Science, The University of Tokyo, Tokyo, Japan. [2] Department of Neurosurgery, The University of Tokyo Hospital, Tokyo, Japan. [3] Department of Data Science, National Center for Global Health and Medicine in Japan, Tokyo, Japan. [4] Leading Center for the Development and Research of Cancer Medicine, Juntendo University, Tokyo, Japan. [5] Department of Pathology, Kyorin University School of Medicine, Tokyo, Japan. ✉email: toudou-nsu@umin.ac.jp

Glioblastoma, the most common and aggressive form of brain tumor in adults, has an extremely poor prognosis, with 5-year survival rates of <10% after standard treatment[1]. The standard-of-care treatment for primary glioblastoma has been chemoradiotherapy with 60 Gy total of fractionated irradiation combined with the alkylating agent, temozolomide, after maximum surgical resection. Tumor-treating fields (TTF), which delivers low intensity alternating electric fields to the tumor[2], is currently included in the standard-of-care for newly diagnosed glioblastoma in the United States. The regimen using TTF has been reported to provide a median overall survival of 20.9 months[2], but its recommendation level as a standard-of-care varies among countries outside the US. Glioblastoma invariably recurs and there is no effective treatment at recurrence. Meta-analysis data of phase II studies show a median survival of 5.0 months and a 1-year survival rate of only 14% for patients with glioblastoma after recurrence[3]. Therefore, to improve outcomes in patients with glioblastoma, new therapies that differ from conventional approaches need to be developed[4].

Oncolytic virus therapy is an approach first conceptualized ~70 years ago, in which a replication-competent virus is used to treat cancer[5]. Commonly, viruses are genetically engineered to restrict virus replication to cancer cells, and infected cancer cells are destroyed directly by the virus in the course of viral replication. Herpes simplex virus type 1 (HSV-1) is especially suited for cancer therapy, because: (1) it infects most cell types, so can be applied to a variety of cancer types; (2) a relatively low multiplicity of infection is needed for total cell killing; therefore, high efficacy can be expected in clinical settings; (3) availability of anti-viral drugs means that therapy can be terminated, if necessary; (4) the large genome allows insertion of large and/or multiple transgenes; (5) circulating anti-HSV-1 antibody does not affect cell-to-cell spread of the virus; therefore, efficacy is not inhibited in seropositive patients and dosing can be repeated without diminishing efficacy[6]. G207 is one of the first oncolytic HSV-1 used in clinical trials[7], with deletions in both copies of the $\gamma34.5$ gene and a *lacZ* insertion inactivating the *ICP6* gene[8]. Intratumoral inoculations with G207 in a syngeneic mouse tumor model of N18 (neuroblastoma) revealed that, in addition to a direct oncolytic activity, the antitumor efficacy is augmented by induction of a systemic and specific antitumor immunity associated with a longstanding elevation of a cytotoxic T lymphocyte activity against tumor cells[9]. G207 has been shown to be safe in patients with malignant glioma, including children, in several clinical trials[7,10,11]. However, there seems to be room for improvement in efficacy with G207.

G47Δ, a third-generation, triple-mutated oncolytic HSV-1, was created by introducing a further deletion within the *α47* gene of the G207 genome[12]. The deletion of the overlapping *US11* promoter results in placement of the late *US11* gene under the control of the immediate-early *α47* promoter, which leads to partial recovery of the deleted $\gamma34.5$ functions, and results in enhanced replication in tumor cells[13]. The main function of the *α47* gene is to allow the virus to escape from immune surveillance by downregulating the major histocompatibility complex (MHC) class I expression in infected host cells through binding the *α47* gene product to the transporter associated with antigen presentation[14]. The *α47* deletion in G47Δ causes a further attenuation of the virus in normal cells, but enhances the stimulation of antitumor immune responses[12]. In fact, G47Δ demonstrated higher cytocidal activity and greater antitumor efficacy than G207 both in vitro and in vivo, while retaining high safety characteristics[12]. In addition to glioma, G47Δ has also shown efficacy in a wide range of solid tumors[5,15–24]. A phase 1 study was completed in patients with castration-resistant prostate cancer, in which $3 \times 10^8$ pfu (plaque-forming units) of G47Δ was injected into the prostate using a transrectal ultrasound-guided transperineal technique (UMIN000010463).

Also, a phase 1 study was recently completed in patients with malignant pleural mesothelioma (UMIN000034063, jRCTs033180326), in which $1 \times 10^9$ pfu of G47Δ was injected repeatedly into the pleural cavity (result unpublished), and another phase I trial is ongoing in patients with olfactory neuroblastoma (UMIN000011636, jRCTs033180325). Further, G47Δ has been shown capable of killing cancer stem-like cells derived from glioblastoma patients and inhibiting their self-renewal[25–28].

Here, we report the results of a phase I/II trial using G47Δ in Japanese patients with recurrent or progressive glioblastoma. This single-arm, dose-escalation study primarily aims to assess the safety of repeated stereotactic intratumoral G47Δ administrations at two dose levels. We follow patients enrolled in this study for a long period for any long-term G47Δ-related adverse events in survivors. Secondary objectives are to assess the efficacy of G47Δ in terms of tumor shrinkage assessed by MRI, overall survival, and progression-free survival. We show that G47Δ is safe for treating glioblastoma and that further clinical development is warranted.

## Results

**Study design and patient characteristics.** This phase I/II study was conducted from November 2009 to November 2014 (data cut-off date, November 27, 2014; survival confirmed until March 1, 2022) at the University of Tokyo Hospital and the Institute of Medical Science Hospital of the University of Tokyo. According to the study protocol, it was required to allow a certain period of time between patients and between cohorts to ensure safety. This study consisted of a phase I part and a phase II part. The study protocol for the phase I part was a typical $3 \times 3$ scheme with a 3-fold dose-escalation (Supplementary Fig. 1). The highest dose tested in the phase I part was to be further used in the phase II part. The Independent Data Monitoring Committee (IDMC) decided not to advance to Cohort 3 in the phase I part, and to set the Cohort 2 dose, $1 \times 10^9$ pfu per dose (total dose, $2 \times 10^9$ pfu), as the highest dose in this study. This decision was based mainly on the finding that all three patients in Cohort 2 showed increased frequency of convulsion. In addition, because G47Δ replicates within the tumor, the administered dose is amplified and the actual amount of G47Δ should vary widely among patients. Therefore, the conventional 3-fold dose-escalation scheme often used for chemotherapy was considered unsuitable for this study. Thus, $1 \times 10^9$ pfu per dose (total $2 \times 10^9$ pfu) was the highest dose tested in the phase I part, and this dose was used for the phase II part. Seven additional patients were enrolled in the phase II part and, in November 2014, the completion of this study was approved as the Phase II clinical trial (UMIN000015995) started.

Accordingly, 13 patients who completed treatment between November 2009 and July 2014 (3 patients in Cohort 1, ten patients in Cohort 2 and the phase II part) were included in this study (inclusion/exclusion criteria, Supplementary Table 1). These 13 enrolled patients comprised the safety evaluation (safety analysis set) population as well as the full analysis set (FAS). Patients received stereotactic intratumoral administration of G47Δ followed by a second administration into the same coordinates after 5–14 days. Patients in Cohort 1 received G47Δ at a dose of $3 \times 10^8$ pfu (total $6 \times 10^8$ pfu), which corresponded to the low dose, whereas patients in Cohort 2 and the phase II part received G47Δ at a dose of $1 \times 10^9$ pfu (total $2 \times 10^9$ pfu), which corresponded to the set dose. For each administration, a total volume of 1 mL was equally divided and G47Δ was injected to two different sites (0.5 mL per site) of contrast-enhanced areas within the tumor in all patients.

Patient demographics and clinical characteristics are shown in Table 1. The median age (range) was 46 (35–76) years; eight patients were male, and five patients were female, and five

**Table 1 Baseline demographic and clinical characteristics.**

| # | Age | Tumor Site (R/L) | Extent of surgical resection | RT (Gy) | CTX | Recurrence before G47Δ | Time from initial surgery to last recurrence (months) | KPS (%) | Tumor size (mm); Cross-sectional area (mm²) | IDH1 | MGMT expression | Surgery for first recurrence | RT for first recurrence | CTX for first recurrence | Steroid | Serum HSV antibody positivity |
|---|---|---|---|---|---|---|---|---|---|---|---|---|---|---|---|---|
| 1 | 76 | OL (L) | GTR | 60 | TMZ, IFNβ | First | 7.5 | 70 | 53 × 44; 2332 | wt | + | No | No | TMZ | DEX (2 mg) | Positive |
| 2 | 36 | TL (L) | GTR | 60 | ACNU, IFNβ | Second | 29.7 | 90 | 19 × 19; 361 | mt | − | Yes | No | TMZ | No | Positive |
| 3 | 35 | PL (R) | PR | 60 | TMZ, IFNβ | First | 5.7 | 80 | 51 × 44; 2244 | wt | − | No | No | TMZ + IFNβ | No | Positive |
| 4 | 60 | FL (L) | PR | 60 | TMZ | Second | 13.0 | 70 | 25 × 9; 225 | wt | + | Yes | No | TMZ | No | Positive |
| 5 | 61 | TL (R) | GTR | 60 | TMZ | Second | 24.4 | 90 | 29 × 26; 754 | wt | + | Yes | Cyber knife 30 Gy/5fr | No | No | Negative |
| 6 | 55 | FL (R) | PR | 60 | TMZ | Second | 8.1 | 90 | 28 × 26; 728 | wt | − | No | No | TMZ + IFNβ | No | Positive |
| 7 | 42 | Splenium | PR | 60 | TMZ, BEV | First | 12.0 | 90 | 62 × 34; 2108 | wt | − | No | No | TMZ | No | Negative |
| 8 | 42 | FL (R) | GTR | 60 | TMZ | First | 12.7 | 100 | 33 × 33; 1089 | wt | − | No | No | No | PSL (15 mg) | Positive |
| 9 | 43 | TL (R) | GTR | 60 | TMZ | Second | 9.2 | 80 | 42 × 22; 924 | wt | + | Yes | No | No | BTM (2 mg) | Negative |
| 10 | 50 | FL (R) | Biopsy | 54 | TMZ | Second | 22.4 | 60 | 33 × 24; 792 | wt | − | Yes | No | No | No | Negative |
| 11 | 40 | PL (L) | Biopsy | 54 | ACNU | Second | 90.5 | 90 | 40 × 22; 880 | mt | ++ | Yes | No | TMZ | No | Positive |
| 12 | 62 | TL (L) | GTR | 60 | TMZ | Second | 18.9 | 80 | 58 × 28; 1624 | wt | + | Yes | No | No | No | Positive |
| 13 | 45 | OL (R) | GTR | 60 | TMZ, IFNβ, BCNU | First | 9.5 | 90 | 39 × 21; 819 | wt | + | No | No | No | No | Positive |

Abbreviations: ACNU nimustine; BCNU carmustine; BEV bevacizumab; BTM betamethasone; CTX chemotherapy; DEX dexamethasone; FL frontal lobe; IFNβ interferon-β; HSV herpes simplex virus; KPS Karnofsky Performance Score; mt mutant; OL occipital lobe; PL parietal lobe; PSL prednisolone; RT radiotherapy; TL temporal lobe; TMZ temozolomide; wt wild type.
Extent of surgical resection: Gross total resection (GTR) is defined as >95% of the tumor removed as assessed by the surgeon. PR partial resection.
Serum HSV antibody is judged positive if IgG ≥ 2.0 EIU (enzyme immunoassay unit).
Tumor size is that at the first G47Δ administration.
Immunohistochemistry for MGMT expression: −, <10%; +, ≥10% and <50%; ++, ≥50%.

patients were undergoing the first recurrence and eight patients were undergoing the second recurrence. The median (range) time to last recurrence was 12.7 (5.7–90.5) months and the median (range) cross-sectional area of these recurrent tumors (measured at the largest dimension) was 880.0 (225–2332) mm². Neither IDH mutation nor MGMT promoter methylation status was available at the time of this study. Post-hoc immunohistochemistry for IDH1 mutation (R132H) and MGMT expression using biopsy specimens revealed two patients (15.4%) positive for IDH1 mutation, and seven patients (53.8%) showed positive immunostaining for MGMT.

**Safety**. Regarding safety outcomes (primary endpoint), adverse events occurred in all three patients (100%) in the low dose group

and nine of ten patients (90%) in the set dose group. The most common adverse events occurring up to 90 days after the last G47Δ administration included headache ($n = 8$ [61.5%]), fever ($n = 8$ [61.5%]), and vomiting ($n = 6$ [46.2%]) (Table 2). Among these adverse events, Grade 3 adverse events were only leukopenia, headache, vomiting, neurological disorder, and IX–XI cranial nerve disorder in one patient each. An increase in the frequency of seizures that was observed in all three patients in Cohort 2 was found to be coincidental, since such was not observed in the following seven patients who received the same dose (the phase II part). One Grade 4 adverse event was lymphocyte depletion in Cohort 1, but a causal relationship with G47Δ was ruled out (Supplementary Table 2). Adverse events for which a relationship with G47Δ could not be ruled out were fever

**Table 2 Summary of adverse events occurring up to 90 days after G47Δ ($N = 13$).**

| Event | Grade 1 | Grade 2 | Grade 3 | Grade 4 | Total number of patients (%) |
|---|---|---|---|---|---|
| Constitutional symptoms | | | | | |
| Fever | 7 | 1 | 0 | 0 | 8 (62) |
| Fatigue | 0 | 1 | 0 | 0 | 1 (8) |
| Falling | 1 | 0 | 0 | 0 | 1 (8) |
| Pain | | | | | |
| Wound pain | 2 | 0 | 0 | 0 | 2 (15) |
| Neurology: Head/headache | 6 | 1 | 1 | 0 | 8 (62) |
| Musculoskeletal: Buttock | 0 | 1 | 0 | 0 | 1 (8) |
| Gastrointestinal | | | | | |
| Nausea | 3 | 1 | 0 | 0 | 4 (31) |
| Vomiting | 2 | 3 | 1 | 0 | 6 (46) |
| Anorexia | 3 | 0 | 0 | 0 | 3 (23) |
| Constipation | 1 | 0 | 0 | 0 | 1 (8) |
| Blood/bone marrow | | | | | |
| Hemoglobin | 2 | 0 | 0 | 0 | 2 (15) |
| Leukocytes (total WBC decreased) | 1 | 2 | 1 | 0 | 4 (31) |
| Lymphopenia | 0 | 1 | 0 | 1 | 2 (15) |
| Coagulation | | | | | |
| INR | 1 | 0 | 0 | 0 | 1 (8) |
| Metabolic/Laboratory | | | | | |
| Bilirubin (hyperbilirubinemia) | 1 | 0 | 0 | 0 | 1 (8) |
| GGT (elevated) | 1 | 0 | 0 | 0 | 1 (8) |
| ALT, SGPT (elevated) | 2 | 0 | 0 | 0 | 2 (15) |
| Neurology | | | | | |
| Seizure | 0 | 4 | 0 | 0 | 4 (31) |
| Neuropathy: cranial-CN VI | 1 | 0 | 0 | 0 | 1 (8) |
| Neuropathy: cranial-CN IX–XI | 0 | 0 | 1 | 0 | 1 (8) |
| Tremors | 1 | 0 | 0 | 0 | 1 (8) |
| Neuropathy: motor | 0 | 0 | 1 | 0 | 1 (8) |
| Neuropathy: sensory | 0 | 1 | 0 | 0 | 1 (8) |
| Cardiac general | | | | | |
| Hypertension | 0 | 1 | 0 | 0 | 1 (8) |
| Hypotension | 1 | 0 | 0 | 0 | 1 (8) |
| Cardiac arrhythmia | | | | | |
| Supraventricular and nodal arrhythmia- Sinus bradycardia | 0 | 1 | 0 | 0 | 1 (8) |
| Dermatology/Skin | | | | | |
| Rash/desquamation | 1 | 0 | 0 | 0 | 1 (8) |
| Pruritus/itching | 2 | 0 | 0 | 0 | 2 (15) |
| Hemorrhage/Bleeding | | | | | |
| Hemorrhage with surgery | 3 | 0 | 0 | 0 | 3 (23) |
| Hemorrhage: Genitourinary (urethra) | 1 | 0 | 0 | 0 | 1 (8) |
| Syndromes | | | | | |
| Flu-like syndrome | 1 | 0 | 0 | 0 | 1 (8) |
| Allergy/Immunology | | | | | |
| Allergic rhinitis | 1 | 0 | 0 | 0 | 1 (8) |
| Total | 45 | 18 | 5 | 1 | 12 (92)* |

Abbreviations: ALT alanine aminotransferase; GGT γ-glutamyl transpeptidase; SGPT serum glutamic pyruvic transaminase; WBC white blood cell.
Values in Grade 1 to Grade 4 columns represent numbers of patients. The highest-grade adverse event is counted for each patient.
* Total number of patients (percentage) with any adverse events.

(12 events in seven patients), pain-head/headache (nine events in seven patients), vomiting (six events in four patients), convulsion (six events in three patients), nausea and intratumoral hemorrhage (three events in three patients), wound pain (four events in two patients), decreased leukocyte count and anorexia (two events in two patients), decreased hemoglobin, tremor, and VI cranial nerve disorder (one event each in one patient) (Supplementary Table 2). Fever did not seem to correlate with the seropositivity to HSV at the time of G47Δ treatment. Cranial nerve VI disorder was temporary diplopia on day 2 of the first G47Δ administration that disappeared the next day. The hemorrhages were all evidenced within 1 day of G47Δ administration by CT scan, and were all small and asymptomatic, likely caused by biopsies. Although, according to the protocol, "expected" events were excluded from adverse events, an increase in surrounding edema was observed in six patients.

**Viral shedding.** Quantitative PCR of G47Δ DNA in blood, urine and saliva on the day after the first dose, the day after the second dose, and 7 days after the first dose were consistently negative (Supplementary Table 3). Further, viral culture of the serum was also negative at each time point of testing.

**Efficacy.** The course of outcomes for individual patients is shown in Table 3. Because enlargement of the target contrast enhanced lesion was observed in most patients immediately after G47Δ administration, according to the study protocol that adopted the WHO-Response Evaluation Criteria, either complete response (CR) or partial response (PR) was seen in no patients (0.0%), whereas stable disease (SD) was seen in two patients (15.4%), and

progressive disease (PD) was seen in 11 patients (84.6%). However, the actual best overall response (BOR) observed on MRI during the 2 year observation period, an exploratory analysis, was CR in one patient (7.7%), PR in one patient (7.7%), SD in s patients (46.2%), and PD in five patients (38.5%). CR occurred in a patient who received the low dose of G47Δ.

**Treatment outcomes.** At the end of the 2-year observation period from the last G47Δ administration, 10 of 13 patients died (Table 3). One patient that showed PR survives for >11 years after G47Δ therapy (as of March 1, 2022). This patient survives without further recurrence and without developing any long-term G47Δ-related adverse events including autoimmune disease in the central nervous system, a theoretically anticipated side effect. Among 12 deaths, two patients achieved long-term survival for periods of 46.7 months and 47.0 months, respectively. None of the three long-term survivors was IDH1 (R132H) mutated.

The median overall survival (OS) from the initial surgery (initial diagnosis) was 30.5 (95% CI 19.2–52.7) months. The median overall survival (OS) from the last G47Δ administration was 7.3 (95%CI 6.2–15.2) months (Fig. 1), with a 1-year survival rate of 38.5 (95%CI 13.9–68.4) % and 2-year survival rate of 23.1 (95%CI: 5.0–53.8)%. The median progression-free survival (PFS) from the last G47Δ administration was 8 (95%CI 7–34) days, mainly due to immediate enlargement of the contrast-enhanced area of the target lesion, although PFS for individual patients extended up to 382 days in Patient #3.

**Imaging and histological findings.** Two common features were observed on MRI after G47Δ administration: (1) clearing of

**Table 3 Summary of patient outcomes.**

| # | Cohort (Dose, pfu) | Age | First and second interval (days) | Response during protocol | Time to progression (days) | Best response | Overall survival (months) | | Final outcome |
|---|---|---|---|---|---|---|---|---|---|
| | | | | | | | After first surgery | After G47Δ therapy | |
| 1 | Cohort 1 ($3.0 \times 10^8$) | 76 | 14 | PD | 7 | PD | 11.1 | 3.6 | Dead (tumor progression) |
| 2 | Cohort 1 ($3.0 \times 10^8$) | 36 | 14 | PD | 27 | SD | 37.0 | 7.3 | Dead (tumor progression) |
| 3 | Cohort 1 ($3.0 \times 10^8$) | 35 | 7 | SD | 382 | CR | 52.7 | 47.0 | Dead after improved (tumor progression) |
| 4 | Cohort 2 ($1.0 \times 10^9$) | 60 | 9 | PD | 7 | PR | >156.5* | >143.9* | Stable* |
| 5 | Cohort 2 ($1.0 \times 10^9$) | 61 | 12 | PD | 7 | SD | 31.5 | 7.2 | Dead (tumor progression) |
| 6 | Cohort 2 ($1.0 \times 10^9$) | 55 | 7 | PD | 8 | PD | 23.2 | 15.2 | Dead (tumor progression) |
| 7 | Phase II ($1.0 \times 10^9$) | 42 | 5 | PD | 23 | SD | 15.2 | 3.2 | Dead (tumor progression) |
| 8 | Phase II ($1.0 \times 10^9$) | 42 | 13 | PD | 8 | PD | 19.2 | 6.4 | Dead (tumor progression) |
| 9 | Phase II ($1.0 \times 10^9$) | 43 | 7 | SD | 95 | SD | 55.8 | 46.7 | Dead (tumor progression) |
| 10 | Phase II ($1.0 \times 10^9$) | 50 | 6 | PD | 39 | SD | 29.4 | 6.9 | Dead (tumor progression) |
| 11 | Phase II ($1.0 \times 10^9$) | 40 | 7 | PD | 7 | PD | 96.7 | 6.2 | Dead (tumor progression) |
| 12 | Phase II ($1.0 \times 10^9$) | 62 | 7 | PD | 7 | SD | 30.5 | 11.6 | Dead (tumor progression) |
| 13 | Phase II ($1.0 \times 10^9$) | 45 | 6 | PD | 34 | PD | 23.2 | 13.7 | Dead (tumor progression) |

*As of March 1, 2022.

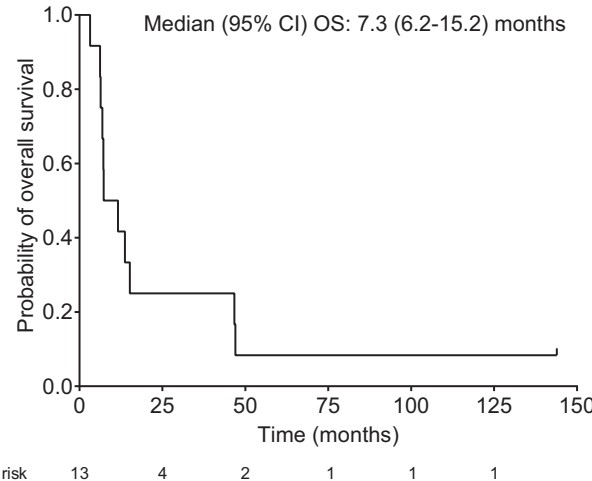

**Fig. 1 Kaplan–Meier curve for overall survival from the last G47Δ administration (FAS population; N = 13).** Source data are provided as a Source Data file.

contrast enhancement at the injection site, and (2) enlargement of the entire enhanced lesion. These characteristic MRI features occurred in most patients and within 14 days (and mostly within 7 days) of the first G47Δ injection (Figs. 2, 3; Supplementary Fig. 2). Biopsy specimens obtained immediately prior to the second injection, 5–14 days after the first injection and from the same coordinates, showed decreased number of tumor cells, possibly due to tumor cell destruction associated with viral replication, infiltration of CD4+ and CD8+ T lymphocytes, and HSV-1 positive immunostaining, likely reflecting G47Δ replication (Figs. 2, 3). The positivity of HSV-1 immunohistochemistry using the same antibody has been shown to correlate with the amount of HSV-1 DNA detected by real-time PCR in in vivo experiments with human malignant glioma xenografts. Lymphocytes tended to infiltrate towards areas of remaining tumor cells rather than areas with HSV-1 positivity. In patient #5, PET imaging with C11 methionine (MET) (MET-PET) performed 3 days after the second dose showed decreased MET uptake at the site of G47Δ injection but a substantially increased uptake in a vast area of the right cerebral hemisphere compared with that before treatment (Fig. 3). Because the initial tumor site of this patient that was grossly resected for diagnosis was the right frontal lobe, the areas of high MET uptake were likely the areas with tumor cell infiltration. MRI images performed 7 days after the second dose showed enlargement of the enhanced lesion. However, immediately after initiation of steroid pulse therapy, associated fever subsided. Steroids were continued for 7 days only, after which the enlarged enhanced lesion on MRI returned to pre-G47Δ treatment size, and MET-PET imaging demonstrated the increased uptake was markedly reduced (Fig. 3). Patient #3 showed a gradual decrease in size of the target lesion and eventually achieved a CR, and the lesions that showed contrast enhancement before G47Δ therapy became areas of porencephaly (Fig. 4). In patient #4, the size of the enhanced lesions increased after the second dose followed by a gradual decrease. A portion of the lesion of Patient #4 continued to show contrast enhancement, so this patient was considered PR at 2 years (Fig. 4), but the lesion continues to stay stable for >11 years after G47Δ therapy. The time course of cross-sectional areas of target tumors for individual patients is shown in Supplementary Fig. 3.

Two (Patients #11 and #13) of the 12 patients who died underwent autopsy. Autopsy brain tumor specimens showed viable glioblastoma cells on hematoxylin and eosin staining, and

persistent intratumoral infiltration of CD4+ and CD8+ T lymphocytes that first became apparent in the early post-dose period (Fig. 5). HSV-1 immunostaining was negative in the brain tumors at autopsy.

**Changes in serum anti HSV-1 antibodies and blood CD4/CD8 ratio.** Seroconversion was observed for HSV-1 antibody (IgG) 1 week after G47Δ treatment in all four patients who were seronegative for HSV-1 (IgG < 2.0) before treatment. In the other 9 patients, IgG antibody titers tended to increase until 2–3 months after G47Δ administration and then decreased (Table 4). Supplementary Table 4 summarizes white blood cell, CD4+ and CD8+ T lymphocyte counts and CD4/CD8 ratio at baseline and on days 30 and 60. Figure 6 shows blood CD4/CD8 ratio measured before G47Δ and 1, 3, 12, 24, and 36 months after the last G47Δ administration. Patient #4, a long-term survivor (>11 years), maintained a high CD4/CD8 ratio.

## Discussion

This study sought to primarily determine the safety of G47Δ in patients with recurrent or progressive glioblastoma. The treatment of repeated intratumoral injections with G47Δ, up to $1 \times 10^9$ pfu/dose, was deemed safe with few severe (Grade ≥ 3) adverse events. Adverse events clearly or likely related to G47Δ were all Grade ≤ 2. HSV-1 is neurotropic and the most common cause of sporadic viral encephalitis in humans, with severe manifestations and a high mortality rate. Therefore, the largest safety concern of oncolytic HSV-1 is naturally its potential to cause encephalitis, especially when injected into the brain. Although T-Vec (talimogene laherparepvec), a second generation oncolytic HSV-1 that has mutations in the γ34.5 and α47 genes and expressing GM-CSF, has been approved as the first oncolytic virus drug in the US and Europe for malignant melanoma[29], preclinical studies have shown that T-Vec (expressing mouse GM-CSF) exhibited toxicity when injected into the brain of BALB/c mice at $1 \times 10^5$ pfu/dose (https://www.fda.gov/vaccines-blood-biologics/cellular-gene-therapy-products/imlygic; Supporting Documents [Approval History, Letters, Reviews, and Related Documents—IMLYGIC]). Therefore, it should be stressed that G47Δ at a concentration of $1 \times 10^9$ pfu/mL and 1 mL/dose, higher than T-Vec used for melanoma patients in clinic ($1 \times 10^8$ pfu/mL and up to 4 mL/dose), showed a favorable safety profile when used in the brain. G207, double mutated in the γ34.5 and ICP6 genes, has proved to be safe when used in the brain in several phase I clinical trials[7,10,11], but preclinical studies have demonstrated that G47Δ exhibits much higher efficacy than G207[12,15]. Therefore, the achievement of both efficacy and safety likely resulted from the careful designing of the G47Δ genome to create mutations effectively in the three viral genes, γ34.5, ICP6 and α47.

The study protocol for the phase I part was a 3 × 3 scheme, with a 3-fold dose escalation similar to the first G207 trial protocol for malignant glioma[7]. The IDMC decisioned to set the second cohort dose ($1 \times 10^9$ pfu/dose) of the phase I part as the highest dose of this study and to use this dose to proceed to the phase II part. Maximum tolerated dose (MTD) is generally defined as the highest dose of a drug that does not cause unacceptable side effects. Dose-limiting toxicity was not observed in this study, so, by definition, MTD was not determined in this study. One of the questions raised by the IDMC was whether the concept of MTD, which is generally equivalent to the optimal dose in the development of chemotherapy drugs, can be applied to development of oncolytic HSV-1. A 3 × 3 scheme has been used for oncolytic viruses[30], but, at the time of this study, the IDMC raised a question regarding the significance of the 3-fold dose-escalation scheme. In fact, in preclinical studies, G47Δ was

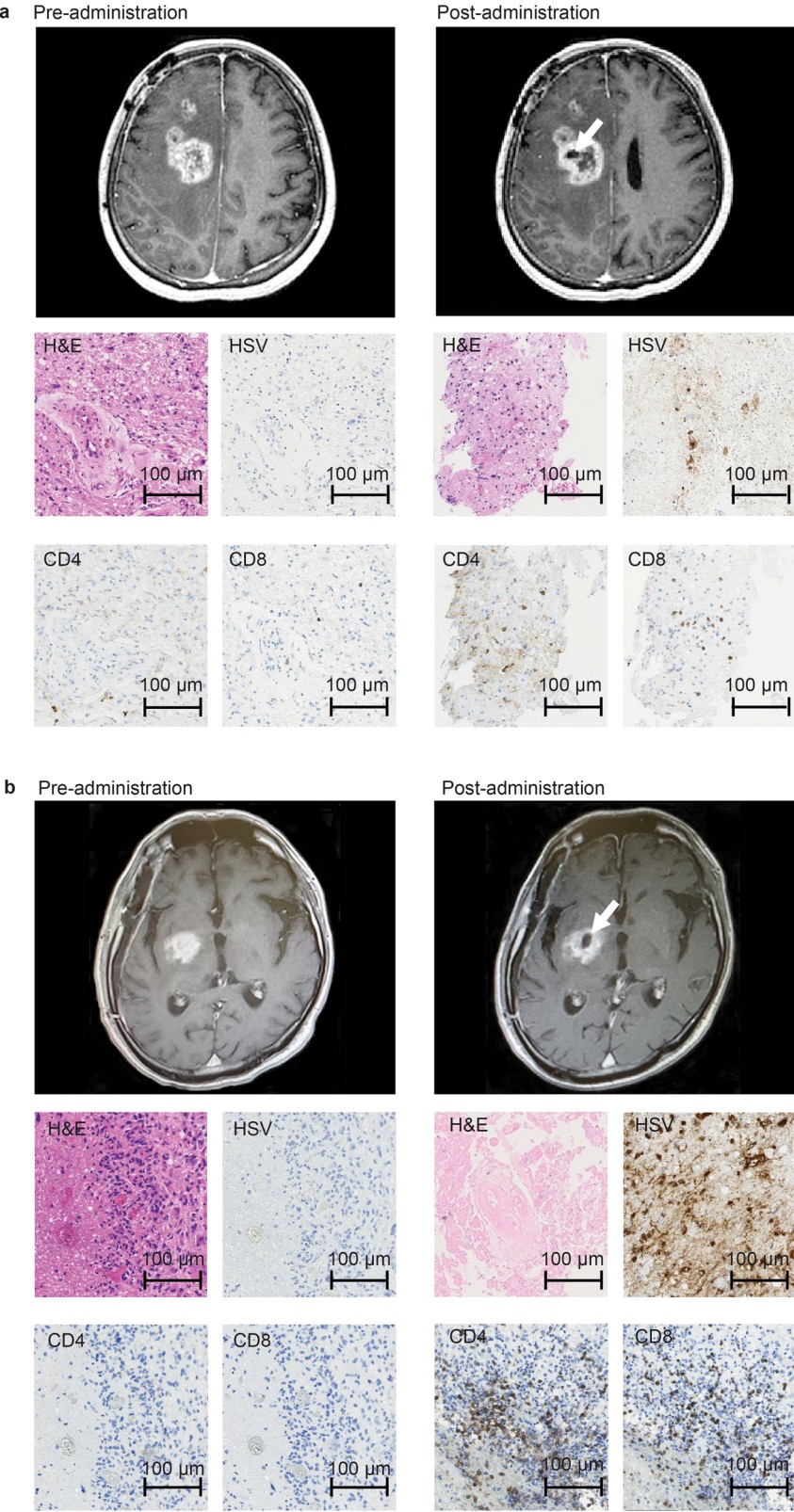

**Fig. 2 MRI and histology studies in two patients (a: Patient #6 and b: Patient #10).** MRI showed enlargement of the entire enhanced lesion with clearing of contrast enhancement at the injection site (arrows) post-G47Δ administration. All specimens from pre-administration were negative for HSV-1 immunostaining. Second biopsy specimens obtained before the second injection from the same coordinates of the first G47Δ injection 7 days (**a**) and 6 days (**b**) post-administration showed decreased number of tumor cells, possibly due to tumor cell destruction associated with viral replication (H&E) and positive HSV-1 immunostaining. An increase in infiltration of CD4+ and CD8+ T lymphocytes towards remaining tumor cells was observed. Pathology images are representative of four biopsy specimens.

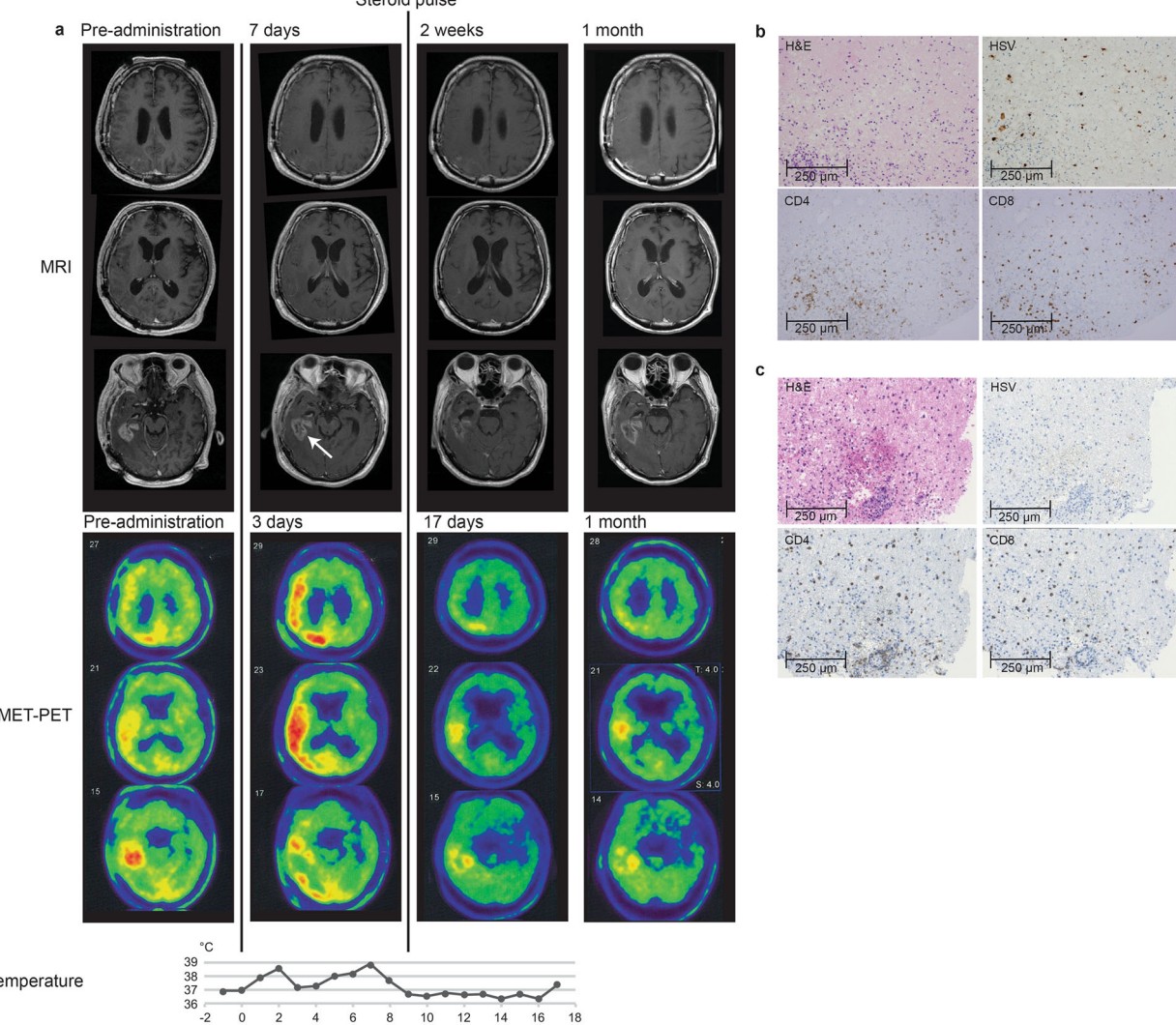

**Fig. 3 MET-PET and histology findings of Patient #5. a** MRI, MET-PET images and the chart of body temperature shown in parallel. MRI showed enlargement of the enhanced legion with clearing of contrast enhancement at the injection site (arrow) 7 days after the second dose of G47Δ. MET-PET performed 3 days after the second dose showed decreased MET uptake at the site of G47Δ injection but a substantially increased uptake in a vast area of the right cerebral hemisphere. Immediately after initiation of 7-day steroid pulse therapy, the enlarged enhanced lesion on MRI returned to pre-G47Δ treatment size, the increased MET uptake markedly reduced, and the fever subsided. The reduced MET uptake event remained so at 1 month. **b** and **c** Biopsy specimen obtained before the second injection from the same coordinates of the first G47Δ injection after 12 days. G47Δ replication was observed by positive HSV-1 immunostaining (**b**). Infiltration of CD4[+] and CD8[+] T lymphocytes was observed in tumor areas around G47Δ (**b**) as well as in tumor areas without G47Δ (**c**). Representative of four biopsy specimens. Source data for body temperature are provided as a Source Data file.

found to show a 100-fold difference in replication capacity depending on the cell type[12], and it was deemed questionable whether the 3-fold dose difference in virus titer is significant. A recent trial with G207, the parental virus of G47Δ, used a 10-fold dose-escalation scheme in children with malignant glioma[31]. DLT has not been seen in the clinical development of G207, and, although $3 \times 10^9$ pfu was the highest dose used in the first phase I trial for malignant glioma, this highest dose was not the dose used for subsequent clinical trials[10,11].

The duration of median overall survival from the initial diagnosis (30.5 months) was longer than expected. Most patients enrolled in this study were relatively young (average 50 years old), possibly reflecting the relatively narrow eligibility criteria. However, a meeting by the Eligibility Assessment Committee took place before every enrollment, and all eligible patients were enrolled sequentially without exception and intentional selection. Post-hoc immunohistochemistry for IDH1 mutation (R132H)

revealed that two patients (15.4%) were IDH1 mutated, but all three long-term survivors were IDH1 wild type. It has been reported that there is no difference in survival time after recurrence of glioblastoma between patients with and without IDH mutation[32]. The adverse events observed with G47Δ in this study were mainly related to immune responses presumably to eliminate explosively replicating G47Δ and towards newly recognized tumor cells. A recombinant polio-rhinovirus chimera (PVSRIPO) was reported to have caused severe brain edema and nervous system disorders with peritumoral inflammation in glioblastoma patients, even causing death in one patient at $3.3 \times 10^8$ 50% tissue-culture infectious doses, and all patients required steroid administration[33]. In contrast, G47Δ caused side effects to a much lesser extent and was generally well tolerated.

On MRI, all patients commonly showed a clearing of contrast enhancement at the injection site and an enlargement of the entire enhanced lesion almost immediately after the first G47Δ

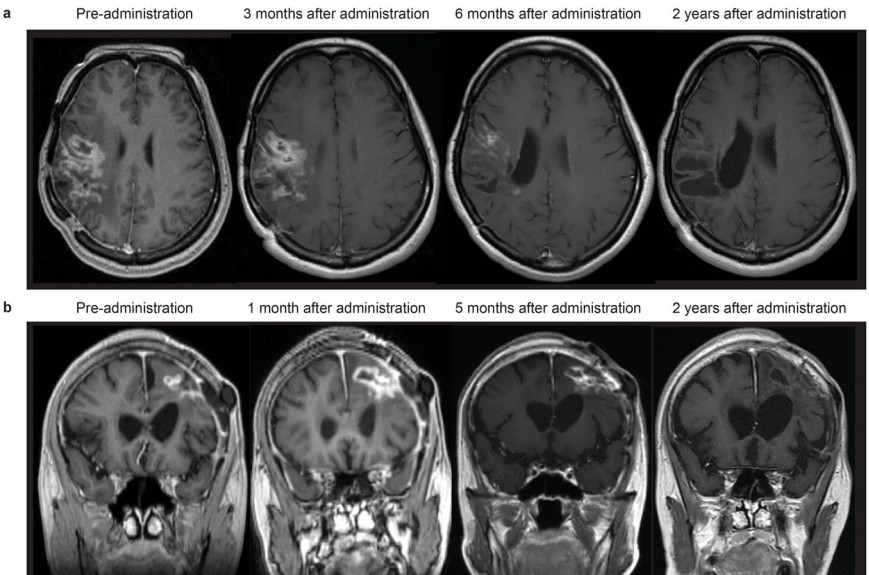

**Fig. 4 Serial MRI findings of two patients who showed responses. a** Patient #3. After receiving two doses of G47Δ ($3 \times 10^8$ pfu/dose), a drastic decrease in size of the target lesion was observed at 6 months and a complete response was observed at 24 months, leaving areas of porencephaly. **b** Patient #4. After receiving two doses of G47Δ ($1 \times 10^9$ pfu/dose), the entire target lesion increased in contrast enhancement size at 1 month, but then gradually decreased at 5 months, and ended with a marked decrease at 2 years. A portion of the lesion continued to show contrast enhancement, so this patient was considered PR. The lesion remains stable and this patient survives for >11 years.

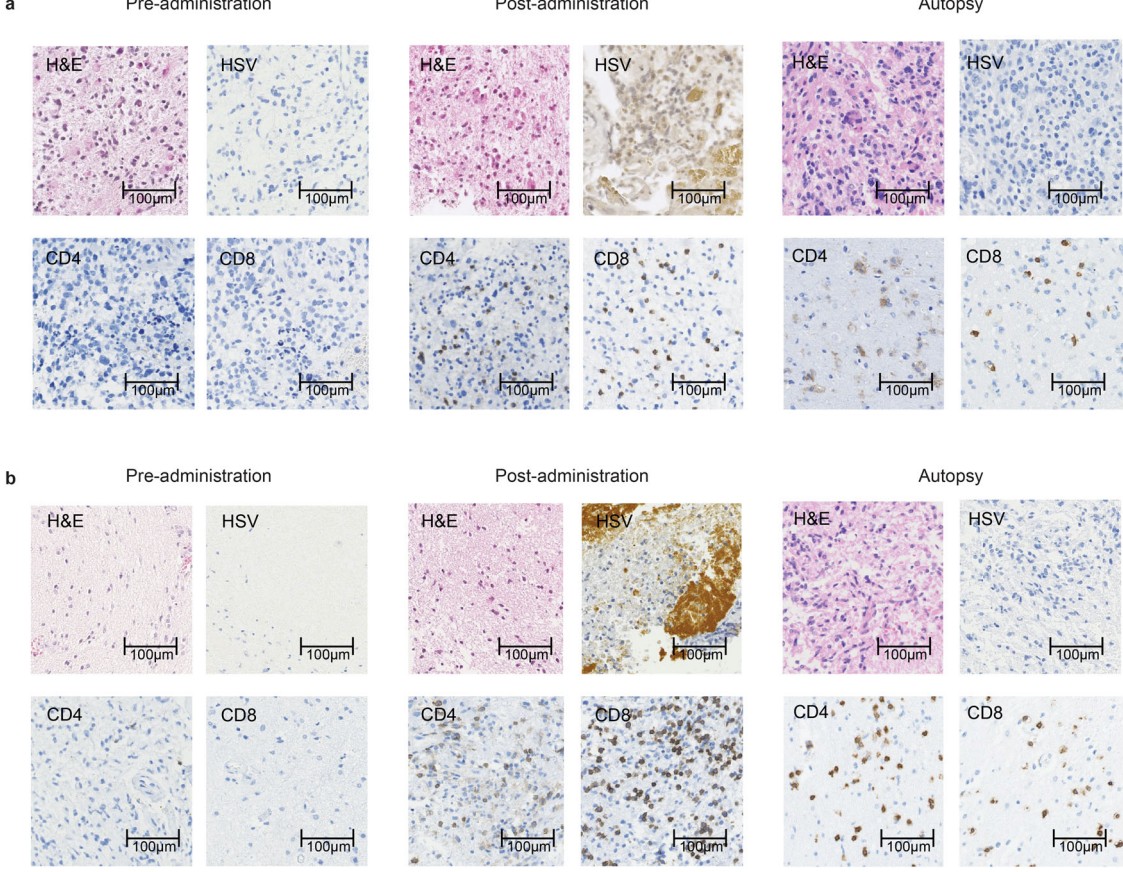

**Fig. 5 Histological findings in autopsy cases. a** Patient #11: 6.2 months after receiving G47Δ and **b** Patient #13: 13.7 months after receiving G47Δ. Both brain tumor specimens from pre-administration showed negative immunostaining for HSV-1, and very scarce CD4+ and CD8+ T lymphocytes. Infiltration of CD4+ and CD8+ T lymphocytes and positive HSV-1 immunostaining were observed after G47Δ administration at the injection site in both cases. Representative of four biopsy specimens. At autopsy, both brain tumor specimens showed viable glioblastoma cells (H&E) together with intratumoral infiltration of CD4+ and CD8+ T lymphocytes. The infiltration of CD4+ and CD8+ T lymphocytes was shown to persist for >13 months. Autopsy brain tumor specimens were negative for HSV-1 immunostaining. Representative of three tissue samples.

**Table 4 Time course of serum anti-HSV-1 antibodies.**

| Patient # | | Pre-admin | 1 week later | 1 month later | 2 months later | 3 months later | 5 months later | 7 months later |
|---|---|---|---|---|---|---|---|---|
| 1 | IgG | 155 | | | | | | |
| | IgM | 1.46 | | | | | | |
| 2 | IgG | 143 | | | 184 | | | |
| | IgM | 0.87 | | | 0.61 | | | |
| 3 | IgG | 26.1 | | 48.6 | | 50.7 | | |
| | IgM | 0.67 | | 0.42 | | 0.51 | | |
| 4 | IgG | 52.4 | | 370 | 465 | 340 | 313 | 230 |
| | IgM | 1.35 | | 1.61 | 2.08 | 2.10 | 1.85 | 1.35 |
| 5 | IgG | <2.0 | 8.3 | 10.5 | 8.1 | | | |
| | IgM | 0.28 | 5.17 | 5.56 | 5.08 | | | |
| 6 | IgG | 40.4 | 90.4 | 46.7 | 83.0 | 80.0 | 52.9 | 54.9 |
| | IgM | 0.71 | 0.73 | 0.70 | 0.62 | 0.61 | 0.63 | 1.04 |
| 7 | IgG | <2.0 | 4.9 | 5.2 | | | | |
| | IgM | 0.36 | 5.26 | 4.88 | | | | |
| 8 | IgG | 101 | 101 | 214 | 262 | | | |
| | IgM | 0.43 | 0.43 | 0.45 | 0.51 | | | |
| 9 | IgG | <2.0 | 15.5 | 14.6 | 17.4 | 19.6 | | |
| | IgM | 0.48 | 4.76 | 5.17 | 2.33 | 2.31 | | |
| 10 | IgG | <2.0 | 10.4 | 11.9 | 16.1 | 22.1 | | |
| | IgM | 0.47 | 3.22 | 4.92 | 2.91 | 1.72 | | |
| 11 | IgG | 80.0 | | 162 | 146 | 181 | 117 | |
| | IgM | 0.15 | | 0.31 | 0.28 | 0.10 | 0.08 | |
| 12 | IgG | 80.0 | | 116 | 137 | 106 | 101 | 93 |
| | IgM | 0.33 | | 0.23 | 0.29 | 0.31 | 0.43 | 0.34 |
| 13 | IgG | 43.9 | | 32.4 | 103 | 79.5 | 108 | 110 |
| | IgM | 0.41 | | 0.53 | 0.51 | 0.57 | 0.57 | 0.65 |

Values are expressed in EIU (enzyme immunoassay unit).

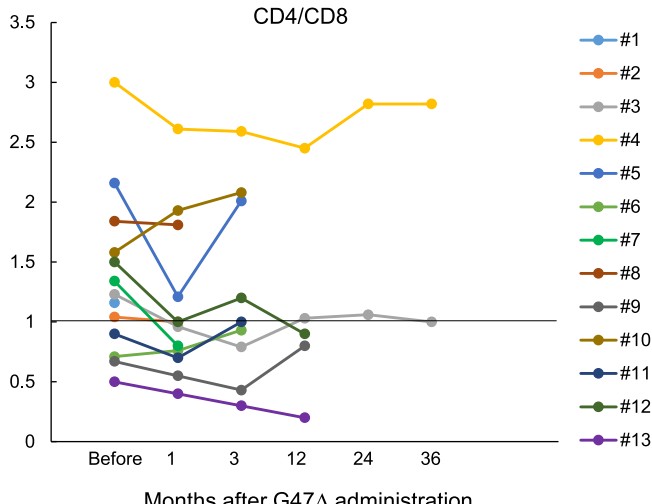

**Fig. 6 Blood CD4/CD8 ratio.** Blood CD4/CD8 ratio was measured before G47Δ and 1, 3, 12, 24, and 36 months after the last G47Δ administration. Patient #4, a long-term survivor, maintained a high CD4/CD8 ratio. Source data are provided as a Source Data file.

administration. This explosion crater-like phenomenon was a fresh observation characterizing the MRI features of glioblastoma treated with G47Δ, and not previously reported with other oncolytic viruses, including G207. The histology of biopsy specimens of the G47Δ injection site obtained 5–14 days later suggests that the clearing on MRI reflects the destruction of tumor cells caused by intense viral replication. Regarding the enlargement of the contrast-enhanced lesion, several lines of evidence point to immune responses as the putative mechanism. Firstly, even for glioblastoma, this enlargement occurred too

soon to be caused by tumor growth. The histology of biopsy specimens showed large numbers of CD4[+] and CD8[+] T lymphocytes seemingly infiltrating into areas with remaining viable tumor cells rather than HSV-1 positive areas, although it is possible that infiltrating lymphocytes could be trafficking towards injected virus or virally-infected cells. Secondly, the enlarged enhanced lesions returned to their original size after 7 days of steroid administration in cases that received steroids. Thirdly, in Patient #5, the high accumulation of MET on PET images between the resected primary tumor and recurrent tumor site, which was observed soon after G47Δ administration, was also resolved after steroid administration. Therefore, the high accumulation observed on MET-PET likely reflected the infiltration of activated lymphocytes towards tumor cells. These findings support the idea that the enlargement of the enhanced lesion on MRI immediately after G47Δ administration is caused by a sudden influx of lymphocytic infiltrates towards the tumor cells. It has been reported that angiogenesis is stimulated by G47Δ in a mouse model[34], and the rapid increase in contrast-enhancement may also reflect the virus-induced increase in vascular permeability[35]. This phenomenon is distinct from the pseudoprogression observed after radiation therapy in terms of both mechanism and timing, so we propose to refer to it as "immunoprogression". Presumably, the infiltration of lymphocytes towards tumor cells occurs, because they come to newly recognize the tumor cells as non-self. Potentially, in the course of the host immune system eliminating G47Δ, antigen-presenting cells process G47Δ bound to destroyed tumor cells and present tumor-associated antigens as non-self targets. It is still uncertain whether and how the emergent lymphocytic infiltrates are related to long-term specific antitumor immunity. MRI results from two patients (#3 and #4) showed that it likely takes 4–5 months for the antitumor immunity to have a therapeutic effect leading to tumor shrinkage. Thus, lymphocytes entering the tumor from a

very early stage of G47Δ therapy may be an unknown subset of lymphocytes with previously unknown roles. Further investigations are necessary to confirm the relationship between "immunoprogression" and specific antitumor immunity.

Steroids have been shown to inhibit the induction of antitumor immunity and to have a negative effect on long-term therapeutic effects in oncolytic HSV-1 therapy[36]. In this study, therefore, we restricted the use of steroids as much as possible. Steroids were used in two patients for only 7 days in both for marked tumor swelling on MRI (Patient #4) and for persistent fever (Patient #5). In these cases, enlargement of the enhanced lesion and a remarkable high accumulation on MET-PET were resolved and the fever recovered immediately after steroid administration. These findings indicate that steroids can effectively counteract the side effects of G47Δ caused by immune responses. Patient #4 remains alive for >11 years after G47Δ treatment, suggesting that 7 days of steroid administration did not adversely affect treatment efficacy.

Several important practical points have been learned from this first-in-human G47Δ study. Firstly, because the "immunoprogression" was commonly observed on MRI in most patients soon after G47Δ administration, conventional response criteria such as WHO and RECIST are not suitable for evaluating oncolytic virus therapy. Instead, this finding led to the formulation of oncolytic virus-specific response criteria referring to immune-related response criteria[37] when designing the subsequent phase II trial. Secondly, host immunity appears to significantly affect the elimination as well as the efficacy of the oncolytic virus[17,25,37–42]. Thirdly, because the extent of viral replication and the strength of immune responses vary among individuals, an effective method for oncolytic HSV-1 therapy may be to repeat the injections based on therapeutic responses. Finally, the long-term efficacy of G47Δ likely depends on the induction of specific antitumor immunity, which may take 4–5 months or more to become apparent; hence, oncolytic virus therapy should be applied at the earlier stage of the disease.

One of the benefits of G47Δ is that it can be combined with other anti-cancer treatments. Given the favorable safety of G47Δ apparent in the present study, it is reasonable to consider that there are few concerns about increased adverse effects when G47Δ is used in combination with other treatments. In the clinic, it is likely that G47Δ will be used in addition to the Stupp protocol when G47Δ becomes widely available for malignant glioma patients. Some promising results have been reported on the therapeutic effects of combining G47Δ and immune checkpoint modulators[43–46]. The results of preclinical data indicate that G47Δ decreases regulatory T cells while recruiting other lymphocytes[47]. Indeed, in a study using various syngeneic murine subcutaneous tumor models, G47Δ and systemic anti-CTLA-4 antibody administration enhanced antitumor activity with upregulation of gene signatures related to inflammation, lymphoid lineage, and T cell activation suggesting an increase in immune susceptibility[47].

In summary, this study proved the safety of G47Δ when administered intratumorally at up to $1 \times 10^9$ pfu/dose for two doses within 14 days. Side effects were mainly caused by immune responses which, when necessary, could be effectively counteracted by a 7-day steroid administration. G47Δ caused immediate infiltration of lymphocytes that seemingly directed towards tumor cells, which was reflected on image studies with features characteristic to G47Δ therapy. Long-term survival (>46 months) was observed in 3 of 13 patients, which may be due to the delayed effect of G47Δ via antitumor immunity. The results justified and formed the basis of the phase II clinical trial in glioblastoma patients, including repeated G47Δ administrations into different coordinates of the tumor and adopting of response criteria specifically designed for oncolytic HSV-1.

## Methods

**Study design and participants**. This was a phase I/II, single-arm study of G47Δ in Japanese patients with glioblastoma. This study consisted of a phase I part and a phase II part. The study protocol for the phase I part was a typical $3 \times 3$ scheme (Supplementary Fig. 1). The phase I part included a 3-fold dose-escalation scheme similar to the first G207 trial protocol for malignant glioma[7]. The highest dose tested in the phase I part was to be further used in the phase II part. The primary objective of this study was to assess safety while efficacy assessment was the secondary objective. The study was registered at the UMIN-CTR Clinical Trial Registry (UMIN000002661) on October 23, 2009. The first patient was registered on November 6, 2009, and the last patient on June 30, 2014. Eligible patients were 18 years or older who had undergone prior surgery at the time of initial or recurrent disease with (i) a pathologically confirmed diagnosis of glioblastoma not responsive to radiation therapy, (ii) MRI-enhanced lesions ≥10 mm, (iii) Karnofsky Performance Scale (KPS) score >60%, (iv) no abnormalities that meet the exclusion criteria and normal major organ function. Patients could be enrolled regardless of prior chemotherapy treatment status. Concomitant usage of antitumor treatment other than G47Δ was not permitted during the 2-year observation period or until the patient is judged PD. Patients were excluded if G47Δ administration would be required to the ventricle (or via the ventricle), brainstem, or posterior fossa, or if any of the following were present: (i) multiple (≥2) glioblastoma lesions, (ii) subependymal and subarachnoid dissemination, (iii) allergy to anti-HSV drugs (acyclovir). A full listing of inclusion and exclusion criteria is included in Supplementary Table 1.

**G47Δ preparation**. A clinical-grade G47Δ product was prepared at the Therapeutic Vectors Development Center of the Institute of Medical Science, the University of Tokyo in accordance with Good Manufacturing Practice. Master cell bank was generated from WHO Vero cells (Seed lot 10-87, RIKEN, Ibaraki, Japan). Generated and purified G47Δ was resuspended in phosphate buffer saline (PBS) containing 10% glycerin, dispensed into cryotubes, and stored at −80 °C. A tube of clinical lot G47Δ was thawed prior to each administration and adjusted to $1 \times 10^9$ pfu/mL in a Biosafety cabinet. Stability tests of the clinical-grade G47Δ were performed in advance to clinical usage by periodic sampling of the product to confirm the stability of virus titer.

**Dose-escalation scheme**. Three cohorts of three patients each were planned to determine the maximum dose of G47Δ (the phase I part). The $3 \times 3$ scheme is shown in Supplementary Fig. 1. An Independent Data Monitoring Committee (IDMC) was to be held at least 14 days after the second dose of the last patient in each cohort. Patients in the three cohorts were to receive two administrations of $3 \times 10^8$ pfu, $1 \times 10^9$ pfu, and $3 \times 10^9$ pfu (total $6 \times 10^8$ pfu, $2 \times 10^9$ pfu, and $6 \times 10^9$ pfu), respectively. After the second cohort, the IDMC decided not to proceed to the third cohort and to set the second cohort dose as the highest dose in this study, mainly because all three patients in the second cohort showed increased frequency of convulsions, which were later found to be coincidental. Further, because G47Δ replicates and amplifies after administration, the actual amount of G47Δ should vary widely among patients. Therefore, the IDMC decided to use $1 \times 10^9$ pfu/dose (total $2 \times 10^9$ pfu), as the set dose to proceed to the phase II part of the study. DLT was not observed in the phase I part, so this set dose was not MTD by definition.

**G47Δ intratumoral administration**. G47Δ was administered directly into the tumor by stereotactic surgery. Each cohort dose was diluted with 10% glycerin PBS to a total volume of 1 mL and, according to the study protocol, divided equally into 2–5 target sites within the tumor. G47Δ, filled into a 1 mL syringe, was administered manually and slowly into the tumor at ~0.2 mL/min using the Biopsy/Injection Needle (MES-CG07-200-01, Mizuho, Tokyo, Japan). After each injection, the biopsy/injection needle was kept in place for 5 min before being pulled out from the brain to avoid leakage. The second dose was given using the same burr hole and to exactly the same coordinates 5–14 days after the first dose.

**Endpoints**. The primary endpoint was safety of G47Δ, assessed by the type, frequency, and severity of adverse events. The frequency of adverse events up to 90 days after the second dose of G47Δ was determined in all treated patients using the National Cancer Institute (NCI) Common Terminology Criteria for Adverse Events (NCI-CTCAE) version 3.0. Relationship of adverse events to G47Δ was assessed on a 5-point scale. Patients also underwent laboratory testing for blood chemistry, coagulation, and lymphocyte fraction (CD4$^+$/CD8$^+$ count and ratio). The assessment of viral shedding was performed according to the study protocol. Quantitative PCR was used to detect G47Δ DNA in blood, saliva and urine collected on the day after the first administration of G47Δ, the day after the second administration, and 7 days after the second administration. In addition, viral culture was performed with blood samples.

Secondary endpoints included tumor response in the 90-day observation period after G47Δ administration. Response was evaluated using WHO-Response Evaluation Criteria based on MRI examination[48]. OS was defined as the period

from the first surgery until the date of death with the date of final survival confirmation censored for survivors. PFS was defined as the period from the second G47Δ dose until progression, which included both PD based on imaging and clinical exacerbations of pathogenesis. Surviving patients who are not considered to have disease progression were censored on the last day of clinical confirmation of the absence of disease progression (date of last PFS). The response rate was defined as the proportion of patients with either CR or PR among eligible patients with measurable disease.

In addition to MRI imaging, PET imaging with C11 methionine (MET) (MET-PET) was conducted in select patients as exploratory analyses. Frequent MET-PET imaging studies were performed only in Patient #5.

**Immunohistochemistry**. IDH1 mutation status and MGMT expression was tested post-hoc by immunohistochemistry of biopsy specimens obtained immediately prior to the first G47Δ injection. Formalin-fixed, paraffin-embedded tissue sections of 3-μm thickness were deparaffinized in xylene and rehydrated in a series of graded alcohols. Antigen retrieval was performed with citrate buffer pH6 for CD4, CD8, and MGM and EDTA buffer pH9 for HSV.

Endogenous peroxidase was blocked by incubation with 3% hydrogen peroxidase for 5 min at room temperature. Then sections were incubated with the following primary antibodies: CD4 (clone EPR6855, dilution 1:250, Abcam); CD8 (clone SP16, dilution 1:100, Abcam); HSV (polyclonal, ready-to-use, Gene Tex); IDH1 R132H (clone H09, dilution 1:100, Dianova); MGMT (clone MT3.1, dilution 1:100, Abcam) for 30 min at 37 °C, respectively. After washing, sections were treated with secondary antibody (Envision + Dual Link System-HRP, Dako) for 30 min at 37 °C. To visualize the antigen-antibody complex, ImmPACT DAB substrate kit (Vector Laboratories) was used, and then sections were counterstained with hematoxylin. Supplementary Table 5 provides details on primary antibodies and antigen retrieval.

**Ethics**. This study protocol was approved by the Ministry of Health, Labour and Welfare of Japan (MHLW) on May 11, 2009 for the University of Tokyo Hospital and on March 22, 2012 for the Institute of Medical Science Hospital. This study was also reviewed and approved by the Gene Therapy Clinical Research Review Board of the participating institutions and conducted in compliance with the following ethical standards and regulations: World Medical Association Declaration of Helsinki (http://www.med.or.jp/wma/); Ministerial ordinance on Good Clinical Practice (GCPs) (MHW ordinance No. 28, 1997: http://law.e-gov.go.jp/htmldata/H09/H09F03601000028.html); Application for Class 1 Use Regulations Based on the Law Concerning the Ensuring Biological Diversity through the Regulatory of the Use of Genetically Modified Organisms; and the Guidelines for Clinical Research on Gene Therapy (http://www.mhlw.go.jp/general/seido/kousei/i-kenkyu/idenshi/0504sisin.html).

Prior to enrollment, informed consent was obtained via a written informed consent form approved by MHLW and the Gene Therapy Clinical Research Review Board and provided to the patient by the treating physician along with a thorough explanation of the treatment procedure.

**Statistical analysis**. Initially, three cohorts of three patients each (nine patients total) were included in determining the safe set dose (phase I part), with an additional 12 patients at the set dose or MTD (phase II part) to reach a target sample size of 21 patients (maximum 30 patients). After the second cohort of the phase I part (six patients), the IDMC decided to use $1 \times 10^9$ pfu/dose (total $2 \times 10^9$ pfu) as the set dose to proceed to the phase II part of this study. Seven additional patients were treated (13 patients total) before the IDMC approved the completion of this study due to the start of Phase II clinical trial. Numerical data on the frequency of different adverse events were tabulated, but no statistical analyses were performed. Overall survival was calculated from the first surgery and from the last administration of G47Δ until the date of death or last follow-up. Progression-free survival was calculated from the date of the last administration of G47Δ until date of progression or period up to the earlier date of death from any cause. The Kaplan–Meier method was applied to describe the distribution of survival time. All data analyses were done with IBM SPSS Statistics version 22 software (IBM Corporation, Somers, USA).

**Reporting summary**. Further information on research design is available in the Nature Research Reporting Summary linked to this article.

## Data availability

All requests for raw and analyzed data will be reviewed and need approval by the Institute of Medical Science Hospital, the University of Tokyo, and the University of Tokyo Hospital. Patient-related data not included in the paper were generated as part of a clinical trial and are subject to patient confidentiality. All data shared will be de-identified. Requests should be made to the corresponding author. The study protocol is available as Supplementary Note 1 in the Supplementary Information file. The remaining data are available within the Article, Supplementary Information, or Source Data file. Source data are provided with this paper.

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

## Acknowledgements

We thank Drs. Shoji Tsuji, Kazuhiro Nomura, Keiichi Nakagawa, and Hiroshi Oyama, the members of the Independent Data Monitoring Committee. We also thank Drs. Ryozo Nagai, Namie Yamada, Hisashi Koike, and the staffs of the Translational Research Center of the University of Tokyo Hospital, Drs. Takaaki Kirino, Nobuhito Saito and the staffs of the Department of Neurosurgery of the University of Tokyo Hospital, the members of the Brain Tumor Therapy Research Unit of the Center for Integrated Brain Medical Science, the University of Tokyo, Drs. Motokazu Ito, Miwako Iwai and the members of the Division of Innovative Cancer Therapy, and the staffs of the Center for Translational Research, the Institute of Medical Science, the University of Tokyo, and Ms. Yuko Ouchi of The University of Tokyo Health Service Center for their support and assistance in carrying out this clinical study. We further thank Drs. Kiyoshi Miyagawa, Nobutaka Kawahara, Kensuke Kawai, Kyosuke Kamada, Hirofumi Nakatomi for participating in the Eligibility Assessment Committee. This work was supported in part by grants to T.T. from the Japanese Ministry of Education, Culture, Sports, Science and Technology (MEXT) through "the Research Promotion for Innovative Therapies against Cancer", "the Coordination, Support and Training Program for Translational Research", and "Translational Research Network Program", and by The University of Tokyo Hospital. We thank Yoshiko Okamoto, PhD, and Mark Snape, MBBS, of inScience Communications, Springer Healthcare, for helping us write the first draft of the manuscript. This medical writing assistance was funded by Division of Innovative Cancer Therapy, The Institute of Medical Science, The University of Tokyo.

## Author contributions

T.T. was the principal investigator of the study. T.T. performed supervision, investigation, data curation, data analysis, interpretation, and writing and is responsible for methodology and funding acquisition. Y.I and M.T. performed investigation and data curation. M.T. also performed data analysis and writing. H.O. is responsible for data curation and statistical analysis. J.S. is responsible for pathological analysis.

## Competing interests

T.T. owns the patent right for G47Δ in Japan ("Viral vectors and their use in therapeutic methods." 4212897). All other authors have no conflict of interest to declare.
