## [Peer Review File · Nature Communications]

Reviewers' Comments:

Reviewer #1:

Remarks to the Author:

This is a revised manuscript which has been transferred to Nature Communications. Many of my issues remain, and although the authors attempt to the best of their abilities to answer queries, there are fundamental problems with the design and interpretation of the results. For example, the explanation of what an MTD means and how it was used is interesting at best. The authors well know that the only goal of a phase 1 study is TO DETERMINE safety and therefore MTD for a future phase 2/3 trial. This was not done. The overinterpretation of data based on a poorly designed phase 1 trial which led to Japanese approval of a product which has not been proven to be of any benefit is highly problematic and in some ways, reflects poorly on the regulatory agency. Approval of an agent can only occur in the setting of a controlled trial, like phase 2/3, aimed to determine efficacy.

That aside, the study becomes descriptive and almost like a case report.

Reviewer #2:

Remarks to the Author:

The authors have responded to concerns.

Reviewer #3:

Remarks to the Author:

Todo et al. provide a revised version of their FIH clinical trial. Overall, they have answered all my questions. This is an excellent contribution to the oncology literature.

I still have some concerns about 2 areas

1- Viral replication in the human samples. To answer this question, they performed an animal study to show short term HSV antigen and HSV DAN by PCR in a U87 xenograft model. I remain unconvinced that this experiment shows that the short-term antigen expression they see in humans means that there was viral replication. The alternative explanation is that antigen they detect could just represent the initial bolus of infected tumor cells from the viral injection without much, if any, replication.

2- Increased enhancement. The authors try to argue that increased angiogenesis would not occur so rapidly and therefore it must represent increased inflammation. However, what happens rapidly is increased vascular permeability which would lead to leakage of contrast (as shown in animal models from papers by Balveen Kaur in 2007). The increased permeability also would lead to increased lymphocyte infiltration. The rapid increase in enhancement is initially due to virus-induced increased vascular permeability in tumor which then leads to increased perivascular lymphocytes and ultimately more angiogenesis.

Responses to Reviewers

NCOMMS-22-12917-T revised

A phase I/II study of triple-mutated oncolytic herpes virus G47Δ in patients with progressive glioblastoma

Tomoki Todo, Yasushi Ino, Hiroshi Ohtsu, Junji Shibahara and Minoru Tanaka

We sincerely thank the reviewers for re-reviewing our manuscript. Point-by-point responses to the comments of reviewers are detailed below. Changes to the original manuscript are indicated in red in the “marked” revised manuscript.

Reviewer #1:

This is a revised manuscript which has been transferred to Nature Communications. Many of my issues remain, and although the authors attempt to the best of their abilities to answer queries, there are fundamental problems with the design and interpretation of the results. For example, the explanation of what an MTD means and how it was used is interesting at best. The authors well know that the only goal of a phase 1 study is TO DETERMINE safety and therefore MTD for a future phase 2/3 trial. This was not done. The overinterpretation of data based on a poorly designed phase 1 trial which led to Japanese approval of a product which has not been proven to be of any benefit is highly problematic and in some ways, reflects poorly on the regulatory agency. Approval of an agent can only occur in the setting of a controlled trial, like phase 2/3, aimed to determine efficacy. That aside, the study becomes descriptive and almost like a case report.

G47Δ is an oncolytic virus, a new treatment modality totally different from chemotherapeutic reagents. MTD is defined as the highest dose of a drug that does not cause unacceptable side effects (NCI Dictionary of Cancer Terms). Further, MTD is usually considered as the optimal dose in case of chemotherapy. However, this clinical study as well as other oncolytic virus studies (e.g. G207 studies) show that this concept does not necessary apply to oncolytic viruses. As the reviewer points out, the goal of a phase 1 study is to determine safety. Accordingly, this study did prove that the highest dose tested (1×10^9 pfu/dose) was safe, and this dose was in fact used for the subsequent phase 2 study in which the dose was shown to cause a survival benefit, leading to the recent drug approval. It would be reasonable to consider that the lower the dose that shows efficacy compared with the dose that causes DLT, the wider the therapeutic window, which is more beneficial for practical usage.

Reviewer #2:

The authors have responded to concerns.

We thank the reviewer for the comment.

Reviewer #3:

Todo et al. provide a revised version of their FIH clinical trial. Overall, they have answered all my questions. This is an excellent contribution to the oncology literature.

I still have some concerns about 2 areas

1- Viral replication in the human samples. To answer this question, they performed an animal study to show short term HSV antigen and HSV DAN by PCR in a U87 xenograft model. I remain unconvinced that this experiment shows that the short-term antigen expression they see in humans means that there was viral replication. The alternative explanation is that antigen they detect could just represent the initial bolus of infected tumor cells from the viral injection without much, if any, replication.

In vitro studies show that G47 Δ -infected human tumor cells are destroyed within 24 hours. The in vivo experiment performed for this review showed that G47 Δ virus alone (without infection) cannot be detected by HSV-1 immunohistochemistry, therefore the HSV-1 immuno-positive cells are likely the cells infected with G47 Δ within 24 hours. Hence, it is very unlikely that the HSV-1 immunostaining detected 5 -14 days after G47 Δ injection represent the remainder of the initial bolus of infected cells. However, we modified a part of the statement in the Results as follows:
... and HSV-1 positive immunostaining, likely reflecting G47 Δ replication (page 9, line 14).

2- Increased enhancement. The authors try to argue that increased angiogenesis would not occur so rapidly and therefor it must represent increased inflammation. However, what happens rapidly is increased vascular permeability which would lead to leakage of contrast (as shown in animal models from papers by Balveen Kaur in 2007). The increased permeability also would lead to increased lymphocyte infiltration. The rapid increase in enhancement is initially due to virus-induced increased vascular permeability in tumor which then leads to increased perivascular lymphocytes and ultimately more angiogenesis.

We thank the reviewer for this important comment. We agree that the rapid increase in contrast-enhancement may also reflect the virus-induced increase in vascular permeability. We modified a sentence in the Discussion and added a supporting reference as follows (page 14, line 3-4):

It has been reported that angiogenesis is stimulated by G47 Δ in a mouse model³⁴, and the rapid increase in contrast-enhancement may also reflect the virus-induced increase in vascular permeability (Kurozumi K, et al. 2007).

Kurozumi K, Hardcastle J, Thakur R, Yang M, Christoforidis G, Fulci G, Hochberg FH, Weissleder R, Carson W, Chiocca EA, Kaur B. Effect of tumor microenvironment modulation on the efficacy of oncolytic virus therapy. *J Natl Cancer Inst.* 2007 Dec 5;99(23):1768-81. doi: 10.1093/jnci/djm229.

Editorial requests:

1. According to your guidance in the Author Checklist, we revised and removed “first-in-human” from the title, to comply with your formatting requirements, namely avoiding words such as new/novel/first, when referring to the scientific findings. However, the “first-in-human” is a specific terminology used for clinical trials and does not refer to a scientific finding. When we checked with PubMed, we found quite a few recent papers published in Nature Communications using “First-in-human” in their titles, of which all are papers of clinical trials.
2. Since 10 display items (Figures or Tables) are allowed in the main article, original Figure S4, Figure S5 and Table S4 of supplementary information are now included in the main article as Figure 5, Figure 6 and Table 4, respectively. Because pre-administration serum HSV antibody values originally included in Table 1 are shown also in new Table 4, we deleted the values and just show serum HSV positivity in Table 1 to avoid duplication.